# Reading through the Other's Eyes: The Mystical Foundations of Interreligious Dialogue in Chiara Lubich's *Paradise '49*

**Stefan Tobler**

Dep. de Istorie, Patrimoniu și Teologie Protestantă, Universitatea Lucian Blaga din Sibiu, 550024 Sibiu, Romania; stefan.tobler@ulbsibiu.ro

**Abstract:** The life and spirituality of Chiara Lubich (1920–2008), the founder of the Focolare Movement, is marked by a particular mystical experience in the years 1949 and 1950, which found expression in a text entitled *Paradise '49*. In this mystical imprint—according to the thesis of the following paper—the explanation can be found for the fact that Lubich, starting from a traditionally Catholic milieu, followed a path that brought her into dialogue with representatives of all world religions. In particular, phrases with "to live" and "to be" are examined, which point to an existential understanding of religious truth. Dialogue does not mean relativizing one's own truth, but leads to a deeper understanding.

**Keywords:** mystical spirituality; Chiara Lubich; interreligious dialogue; *Paradise '49*; religious truth

## 1. Introduction

In May 1997, Chiara Lubich (1920–2008) was invited to the Malcolm X Mosque in Harlem by Imam Warith Deen Mohammed (1933–2008), a leading representative of Black Muslims in the United States. There, she gave a speech in front of a large group of believers and shared her spiritual experience. The appearance of a white Christian woman in that symbolic place of worship was triply unusual (Coda 1997). Afterwards, the two religious leaders decided to continue working together with a promise of lasting friendship. Was this a moment of naïve idealism, a symbolic act of religious diplomacy with potentially political ramifications, or perhaps merely an emotionally charged moment which should have little bearing on our understanding of Islamic–Christian relations?

Lubich is known above all in Catholic Church circles (see: Robertson 1978; Gallagher 1997; Fondi and Zanzucchi 2003; Torno 2012; Aretz 2019). She was a person of great charisma, and the founder and long-time president of the Focolare Movement, whose beginnings date to 1943 in the northern Italian city of Trent and which spread worldwide in the decades that followed. Because their brand of spirituality laid emphasis on the role of the laity and especially that of women for the mission of the Church in society, the Focolare were often viewed critically by the Church hierarchy in the period before the Second Vatican Council.[1] Despite enjoying the support of the Archbishop of Trent, Carlo de Ferrari (Abignente 2017), it was only after official recognition by the Vatican in 1962[2] that the door opened for the integration of the movement into the Catholic Church, encouraged not least by the respective Popes, above all John Paul II, who held Lubich in high esteem. Throughout her life, Lubich was a deeply devout Catholic woman closely associated with the Roman Magisterium. Beginning in 1960, however, she engaged in inner-Christian ecumenism, and later decidedly took up interreligious dialogue as well. The beginning of this dialogue was the willingness to meet representatives of other religions in openness and friendship. These encounters were characterized primarily by an exchange of spiritual experiences. Only at a later stage did Lubich take the initiative for scientific symposia, whose character can be described with the expression used by Peter L. Berger as dialogic engagement (Berger and Schweitzer 2011) which includes a willingness to learn from each

other and deepen one's understanding of truth. In 1977, Lubich received the Templeton Prize for Progress in Religion and, in 1994, she became honorary president of the World Conference of Religions for Peace (WCRP), at the initiative of Nikkyo Niwano, founder of the Japanese Buddhist movement Risshō Kōsei-kai.

To speak of dialogue in this context is not wrong, but it does not encompass the entire phenomenon of the Focolare Movement. Adherents of other religions who participate in the movement are not only partners in dialogue, but have also adopted the spirituality of this Catholic woman without giving up their respective identities. Today in Algeria, for instance, the majority of those who consider themselves part of the Focolare Movement are Muslim. This phenomenon calls for closer study. Chiara Lubich has always clearly expressed her Christian identity in her encounters with people of other religions, and never limited her message to the realm of general social commitment. Rather, she was always concerned with the exchange of spiritual perspectives and respected the belief in truth as it is found in every religion. She emphasized a common spiritual path without minimizing the differences (see: Catalano 2010; Niwano 2020; Catalano 2015; Callebaut 2021). Roberto Catalano (2015) compares three recent religious movements (the Gülen Movement, the Focolare Movement, and Risshō Kōsei-kai) and describes some commonalities: the reference to a charismatic founding personality, a communal understanding of faith, and the way holy scriptures are read in relation to the contemporary world. The conviction of one's own truth is not minimized or even excluded, but it is understood in a way that makes dialogue and even a common path possible.

In her last year of active work, in 2003, Lubich expressed her conviction that it is possible to "come to a mutual understanding with all the great religious traditions of humanity", and that she sees this from a Christian point of view founded in the work of the Holy Spirit: " . . . the Holy Spirit may in some way be present and active in all religions, not only in the individual members but also in the inner workings of each religious tradition" (according to Catalano 2015, pp. 45–46). The reference to the Holy Spirit signals that Lubich is not concerned primarily with the comparison of dogmatic teachings or with the search for a common denominator in the shaping of the respective faith. It indicates a belief in a common divine force across religions without minimizing the differences between them or engaging in syncretism.

The present paper contributes to the discussion regarding the ways in which Lubich's spirituality contains thought-provoking conceptions for a theology of religions. Considering the long path leading from the young Chiara Lubich's purely Catholic socialization in Trent to the wide-reaching horizon of her later thought, the question arises: are there elements of her spirituality that offer insight into her transformation and thus make her path understandable?

## 2. The Mysticism Intrinsic to Lubich's Spirituality

The basic contours of this open-minded spirituality had already emerged in the years following 1943[3]. Its essential point of departure is faith in God, who is love, and it is anchored by two distinct notions, one of which can be characterized by the term *unity* and the other with the Christological name *Jesus Forsaken*. Significant in this regard is the title of one of her most important writings: *L'unità e Gesù Abbandonato* (Lubich 1984). Often, however, Lubich's works elaborate upon this core in even greater detail, describing ten or twelve points of her faith that serve as a catechetical introduction to the spirituality of the Focolare Movement and which can thus be understood as a mystagogical approach to incorporating faith into one's everyday life (Lubich 2001, pp. 95–210). In her 2001 book *La dottrina spirituale*, she also uses the designation of mysticism (mistica) for her spirituality. This designation is doubly significant.

On the one hand, it refers to a certain period in Lubich's life from July 1949 to the end of 1950 (and some parts of 1951), which was marked by an extraordinary mystical experience. This period will be described in more detail below as the texts that attest to it form the main source for this paper. On the other hand, it is about the basic mystical

character of the religious experience of those who have embraced Lubich's spirituality. It is a mysticism that is fully experienced at the level of daily life and which also possesses a political dimension (Steinmair-Pösel 2019). Although there are good reasons to refrain from defining mysticism and rather to approach it only narratively (Ruh 1990, vol. 1, pp. 13–15), I have tried to circumscribe the mystical character of Lubich's spirituality in five points in "Ecumenism as a Mysticism of Encounter in Chiara Lubich" (Tobler 2020). Together, and their various constellations, these five points make up the mystical character of Lubich's faith. This description is offered here once more—in English translation and with slight modifications—because it can be helpful for the interpretation of Lubich's texts and perhaps even more generally for mystical texts.

1.  *Experience*. Mysticism is not primarily situated on the plane of reason (although reason is not excluded), but involves the existence of man as a whole, including emotion, imagination, the senses, and the body. The whole human person is involved in the act of faith.

2.  *Immediacy*. The mystic experiences God in a way that sees mediation as an open door to a direct relationship with the divine. To speak of mysticism does not mean to reject the established mediation of the Church through the ministry and the sacraments, nor does it mean to reject the mediation of Sacred Scripture as the external Word that must be listened to. But by all of these means, God speaks directly to the heart, through the Holy Spirit.

3.  *Union*. The encounter with God can be experienced as a unifying one, but—at least in the context of Christian mysticism—it does not imply an absorption into an all-encompassing notion of the divine, because it does not abolish humanity and individuality. The form of this unifying experience can vary and may be experienced as an immersion or ascent, as "rapture" in the sense of a change of place, or as a marital union.

4.  *Transformation*. In this union, the mystic does not remain the same. His existence is now one in and with Christ, and he is a new creature (2 Cor 5:17), an experience foreshadowing the divinization which, according to the Christian faith, is promised in the final fulfillment.

5.  *Light*. The encounter with God always includes an element of revelation, that is, of God making himself known to mankind. This deepening of understanding happens primarily with the help of images, symbols, and narratives, and only secondarily with concepts on an abstract level (Tobler 2020, pp. 211–12).

A sixth point should be added here. Mystical experiences are often communicated in written form, because they are understood as a gift from God which should not remain limited to a single person. Since mystical experiences go beyond traditional religious pathways, ordinary language is often insufficient to express them. This requires us to add another point to the description:

6.  *Language*. A mystical experience can lead to modifications in the use and interpretation of traditional religious expressions; it can also lead to the creation of new and unusual expressions.

Lubich's writings, and thus the language with which she expresses her experience, contain an entire linguistic repertoire distinguishing it from traditional Christian language, such as the Christological expressions "Jesus in our midst" and "Jesus Forsaken". This paper, however, will focus on some phrases not exclusively linked to the Christian tradition, but pertaining to the general nature of her religious experience. In my habilitation thesis from 2002, this feature of Lubich's writing is identified by means of the rather unfortunate term, "hermeneutics of identification" (Tobler 2002, p. 146). However, what was observed there is of interest, and concerns the unusual use of the verbs "to live" and "to be". Lubich uses *vivere* (to live) "in a transitive sense and in word combinations that sound as new and uncommon in Italian as they do in German: to live the word, to live unity, to live love, to live the other, to live Jesus, to live Jesus Forsaken, to live the will of God. Similarly, the

verb "to be" can be used with the same predicates: to be the Word, to be love, to be the other, to be Jesus, to be the will of God." (Tobler 2002, p. 147). Lubich believes that the notion of truth in faith is always "lived" truth, existentially experienced truth, whereby, ultimately, God himself is experienced as the one who "is" and "lives" in the believer. In conclusion: "In the underlying, already implied inversion of the subject, and thus in God taking possession of man, 'life' becomes the site of God's self-revelation, every time again" (Tobler 2002, p. 150).

This existential understanding of faith, which would later open up to interreligious dialogue (Mokrani 2006), was already present from the beginning of her spiritual journey[4], but the years 1949 and 1950 led to a decisive deepening. During this period, she had a unique mystical experience which had a profound impact on her path forward. Lubich herself gave the name *Paradise '49* to this experience and to the texts that attest to it. Until now, these have only been studied to a very limited extent. In the following sections, *Paradise '49* will first be introduced briefly, followed by an exploration of its striking use of the words "to live" and "to be", in order to ask, eventually, whether these insights can help to understand the way interreligious dialogue in the Focolare Movement is conceived.

### 3. The Special Mystical Period—*Paradise '49* as Experience and Text

By 1949, Chiara Lubich's movement had developed a surprisingly large following and was supported by the Archbishop of Trent, Carlo de Ferrari (Abignente 2017), despite the mistrust in parts of the Catholic Church (Callebaut 2017, pp. 257–471). That summer, with some of her companions, she withdrew to a village in the Dolomites. There, they were joined by Igino Giordani (Sorgi 2003; Lo Presti 2021), whom she had met in Rome a year earlier. Giordani was a well-known Catholic writer, co-founder of the Christian Democratic Party after World War II, and a member of the Italian Parliament. In his desire to mold his political principles according to the Christian faith, he sought a close connection with Lubich, whose charisma inspired him. However, his subsequent proposal to bind himself to her in a vow of obedience did not meet with her approval. The reasoning behind her reaction may be encapsulated in Jesus' prayer of Jn 17:21, "that all may be one", reflecting her desire for "lived" unity in mutual love rather than obedience. Accordingly, she suggested to Giordani that, following a Eucharistic celebration, they say a prayer asking Jesus to create the unity that He seeks "upon the nothingness" of the two of them (that is, their total openness before God). This moment has become known in the history of the Movement as the "Pact of Unity" and has already been analyzed in depth (Coda 2012). It was particularly important for Lubich because it inaugurated her period of immersion into a divine dimension during the summer and fall of 1949, and at certain points during the following year. Looking back in 1969, she summarized the experience thus:

> It was truly the religious vision of the universe, the religious vision of the world. That is, the way God sees the world, how God sees things, how God sees creatures, how God sees Paradise. (Lubich 2019, p. 4).

In Lubich's understanding of the Christian faith, communication plays a central role. All things, spiritual and material, should be shared as much as possible, because it is the community that allows people to experience the presence of God in the midst of the world. She did this also in 1949 and 1950, in different forms: in letters to Igino Giordani, in oral communication with the companions with whom she lived, and through notes in the form of a diary.[5] Many texts from this period were thus preserved and excerpts from them fell into the hands of the Holy Office in Rome, prompting reservations about this fledgling movement to grow, and eventually spawning an in-depth investigation that was very painful for Lubich (Abignente 2017, pp. 173–79). Her mystical experiences from 1949 and 1950 were very critically assessed, and she was forbidden to speak about them. For a long time, Lubich even believed these writings to be lost. Only after the opening of the Catholic Church in the context of the Second Vatican Council and the recognition of the movement by the Vatican—first in 1962, then in its present form in 1990—did she begin



to speak publicly about these formative years of her life, and to collect and organize the corresponding writings preserved in copies or transcripts.

It was the theologian Klaus Hemmerle, catholic Bishop of Aachen 1975–1994 and close companion to Lubich (Hagemann 2008), who urged that these texts be made more widely accessible. He was convinced that they imparted a revelatory understanding of faith to the modern world and were therefore significant not only for the personal religious journey of Lubich and her followers, but also for the Church and for general theological reflection. Thus, the *Scuola Abbà* Study Group was born in 1990, initially consisting of a small circle of theologians which then grew in number and expanded in academic discipline.

In collaboration with this study group, Lubich collected and arranged all those texts which could be assigned to that extraordinary mystical experience of the years 1949 and 1950. In repeated joint readings, she confronted the hermeneutical challenge posed by these texts, the question of commonalities and differences with comparable mystical experiences in church history, and the question of the texts' relevance for an understanding of the Christian faith that does not avoid dialogue with society and science. From these years of work in the *Scuola Abbà*, numerous comments and explanations by the author have been collected and added as footnotes to the text. Lubich continued working on this project until the year 2004, when she became seriously ill, leaving behind a completed work of about 250 pages entitled *Paradiso '49*. This work serves as a textus receptus and can be interpreted as such, although this does not diminish the future need for a critical investigation into the origins and history of the text.

*Paradiso '49* has not yet been published in its entirety; therefore, in the following sections, some longer quotations are included to help the reader understand the linguistic style and content of Paradise 49. However, numerous paragraphs and fragments from it were already included in other publications. *Nuova Umanità* was the primary vehicle for this activity[6], and beginning in 1995, members of the *Scuola Abbà* published sections from the work in this journal. Between 2012 and 2021, seven volumes of the book series *Studi della Scuola Abbà* were published[7], dedicated to the historical context of *Paradise '49* or individual topics relating to its contents. A number of the texts from *Nuova Umanità* were published in the journal *Claritas. The Journal of Dialogue and Culture* in English translation launched in 2012 and is now available in open access[8]. Some of the most important of these were finally included in an anthology in 2020, along with new commentary (Mitchell 2020). Particularly noteworthy among these, in addition to Chiara Lubich's original texts, are the analyses of the systematic theologian Piero Coda and the exegete Gérard Rossé, who, as long-time members of the *Scuola Abbà*, are among the best experts of *Paradise '49.*

The author of this paper has the complete manuscript at his disposal and refers directly to it.[9] It contains texts of various literary genres, which arranged as far as possible according to the chronological sequence of events, although exact dating remains uncertain in some cases. The description of the preceding experiences and the first days (July 16–18) was written by Lubich, retrospectively, in 1986 and prefixed to the original notes. The oldest fragment is a letter to Igino Giordani from July 19, that is, three days after the beginning of *Paradise '49*. Letters to Giordani are also, occasionally, included subsequently. Beginning on July 20, the account is recorded in the style of a diary, with retrospective notes dated December 1949 inserted in some places. Descriptions of "intellectual visions"[10] (especially in the early weeks) with an abundance of imagery and symbols alternate with more argumentative texts taking up and modifying classical theological language and aiming to shed new light on the Christian faith. Later texts, especially those from 1950, already show clear traces of conversations held by the author with contemporaries and of comparisons with other dogmatic and spiritual traditions. Occasionally, a commentary on a biblical passage is included and there is a single text written with the aim of publication. The content ranges from intuitions about God as Trinity, the work of salvation in Christ, and the nature and destiny of man, to questions of the Church, and (in the second part) the way of life and structure of the Focolare Movement. Throughout the text, questions and the needs of the contemporary world are always given due weight.

## 4. "To Live" and "To Be" in *Paradise '49*

### 4.1. Introduction

A feature of Lubich's spirituality since her early years is the conviction that religious faith has a particular personal relevance and social topicality when it is a *lived faith*, i.e., consciously incorporated into one's own thoughts and actions. This is particularly evident in her approach to the Holy Scriptures. Her focus was not on intellectual understanding, which can be garnered from study of the Bible or church doctrine, but on openness to the divine, which permeates "the Word" and, thus, enters our lives. This is clearly expressed in letters written between 1944 and 1948 (Lubich 2010). Looking back on those years, Lubich describes the experience of herself and the first group of the Focolare Movement as follows:

> The Word of God entered deeply into us, so much so that it changed our mentality. The same thing also happened to those who had some kind of contact with us. This new mentality that was taking shape manifested itself as a true divine protest against the world's way of thinking, of wanting, of acting. And it brought about a re-evangelization in us. The fact is that every Word, even though expressed in human terms and in different ways, is Word of God. But since God is Love, every Word is charity. We believe that at that time, beneath every Word, we had discovered charity. And when one of these Words settled into our soul, it seemed to us that it was transformed into fire, into flames; it was transformed into love. It could be said that our inner life was all love.

> (*P'49* 2–3.7–8, written in retrospect 1986).

This experience and conviction regarding the Word of God is transferred to the whole realm of faith. Only what is "lived", i.e., experienced existentially, can be considered as truly internalized and understood. Therefore, the unusual use of the verb "to live" in a transitive sense can already be found in Lubich's letters preceding 1949: to live the truth, to live Christ, to live the will of God, etc. In the text of *Paradise '49*, this becomes a recurring feature when the author puts the mystical experience into words. The description of the beginning of this period (16 July 1949) is significant. During the aforementioned "Pact of Unity" with Igino Giordani, Lubich returns to the church after the Eucharistic celebration:

> I felt urged to go back into church. I entered and went before the tabernacle. And there I was about to pray to Jesus-Eucharist and say to him: "Jesus." But I could not. In fact, that Jesus who was in the tabernacle was also here in me, was me too, was me, identified with him. Therefore I could not call out to myself. And there I felt coming spontaneously from my lips the word: "Father." And in that moment I found myself within the bosom of the Father.

> (*P'49* 26)[11].

This narrative begins with the experience of union with Jesus in the Eucharistic communion, but culminates in a complete change of perspective, described as being "in the bosom of the Father". For Piero Coda, one of the foremost experts of Lubich's writings, this experience characterizes the distinction between theology and religious studies. Theology does not have God before it as an object of knowledge (Coda 2000), but owes itself to participation:

> I tend to avoid speaking of knowing God. I believe it risks thinking of God as merely an 'object' outside of us. Instead we can refer to knowing God in God, in order to express the specific participation given to us in Jesus, through the gift of his Spirit, in the knowledge that God has of Godself, and in Godself, of all that exists. This, furthermore, emphasizes that theology is most of all about 'being' or 'dwelling' in God, which naturally leads to and expresses itself in a knowledge corresponding to that condition.

> (Coda in: Mitchell 2020, p. 139)[12].

*Paradise '49* is not a theological text, but an expression of a mystical experience; however, we can locate in it the basic structure of theological knowledge. The extent to which

this is expressed in the unusual use of the verbs "to be" and "to live" will be shown here. In doing so, a distinction must be made between two clearly distinguishable parts of the text.

### 4.2. *The Eschatological Vision—"To Be" in the So-Called Realities*

The corpus of *Paradise '49* contains a number of texts that differ markedly in style. Lubich called them *Realities*; this expression does not refer to the classical dogmatic talk of eschatological realities, such as heaven and hell, but to scenes or images that became apparent to Lubich in those weeks. Depending on how they are counted, they constitute 22 to 25 sections of text, ranging in length from one to four pages each. The first bears the date of 19 August 1949, the last that of 9 November 1949. Before this period, the change of place that occurred at the outset of Lubich's mystical journey, and symbolically expressed as the "bosom of the Father", led to a new "seeing": a seeing recognition of the Father, of the Logos, of the Spirit, of Mary, and thus of man in relation to God. In the *Realities*, this seeing changes and becomes a participation, even an identification. The divine reality is no longer perceived as being outside, but becomes an inner experience. Lubich uses expressions in the first person (singular or plural) when she understands and states something of the divine. The duration of such moments of identification is normally one day. She writes, for example: "With Holy Communion today I became: Jesus Forsaken-Humanity" (*P'49* 486, 26 August 1949) or "Today I am Mary, the Mother of God" (*P'49* 622, 8 September 1949). From the perspective of the first person, statements are then made about creation, the salvific work of Christ, the path of man towards God, the understanding of God as Trinity, and other matters.

Lubich herself wrote a brief introductory section to this clearly delineated phase of her mystical experience and prefaced it as a commentary:

> Here begins a series of "Realities" (as we called them) that we lived while in the Bosom of the Father. They were a prelude, a taste of the glorification that we will have in paradise. I do not remember how these Realties began. The vision of things continued to change, because life in Paradise is not static: it is continuous life. From each Reality another developed, each one complete in itself. We loved, we lived a new Eucharist, and so we always went ahead.[13]

These *Realities* are not easy to interpret. Their language is direct and unfiltered, without regard for theological conventions or for the danger that speaking of God in the first-person singular might be interpreted as hubris. For this reason, they are rarely referenced in the literature on *Paradise '49* and have not yet been interpreted in-depth. The sole exception is a series of essays by the exegete Gérard Rossé, wherein he compares the whole of Lubich's mystical experience to the conception of faith in the Epistle to the Ephesians. In these essays, he cites select passages from the *Realities* (Rossé in: Mitchell 2020, pp. 241–327, esp. 255–258 and 278–279) and interprets them in the light of Christian eschatology: the glorification promised to the believer for his future life in God already radiates into our current earthly reality.

The passages belonging to the *Realities* will not be interpreted in more detail in the further course of this paper. They concern a specific phase of Lubich's mystical experience, which may be of great intellectual interest, but is unique and difficult to put in relation to the broader history of the Focolare Movement. This paper is concerned with an experience of faith, not limited to Lubich and her inner circle or to a specific period of time, but rather one that shaped all of her works and also seems to have played a role in the interreligious dialogue that developed long after 1949.

### 4.3. *A More General Way of Talking about "To Live" and "To Be"*

*Paradise '49* utilizes the verbs "to live" and "to be" in a way which links from Lubich's singular mystical experience to a paradigm of Christian life which can guide those who follow her in her spiritual journey.

It is useful to begin with a statistic for the occurrence of these phrases in the text of *Paradise '49*.

| Reference Word | Number in Combination with *"to live"* | Number in Combination with *"to be"* | Total—*"to live"* and *"to be"* |
|---|---|---|---|
| Jesus (rarely: Christ) | 8 | 32 | 40 |
| Word (rarely: Gospel) | 17 | 18 | 35 |
| Jesus Forsaken | 11 | 11 | 22 |
| Maria | 4 | 11 | 15 |
| God | 2 | 9 | 11 |
| Unity | 6 | - | 6 |
| Fellow man/brother | 4 | 2 | 6 |
| One | - | 3 | 3 |
| Church | - | 3 | 3 |
| Wound | - | 3 | 3 |
| Ideal | 2 | 1 | 3 |
| Love | - | 2 | 2 |
| Present moment | 2 | - | 2 |
| Nothing | 1 | 1 | 2 |

In addition, a single reference of "to live" or "to be" can be found with the following: heaven, will of God, misery of the world, sacrament of love, faith, and life.

A first observation concerns the reference word "Jesus". The great frequency of the phrase "being Jesus" lies in contrast with the few passages where Lubich speaks of living or being "like" Jesus (or "like" Jesus Forsaken). This indicates that she does not have in mind an *imitatio Christi*, in which man rationally and deliberately follows the model of Jesus, but that she refers to the experience of union and transformation, as set forth above in the foundational mystical element of Lubich's faith.

It is also striking that there is no significant difference in content between the use of "to live" and "to be". This is most evident in relation to the words "Jesus Forsaken" and "Word", where the number of respective occurrences is equal and where "living" or "being" can be used interchangeably. Two passages from different months show this:

> To live the reality of the marriage of my Soul with the Word: "Love", whom I saw in Paradise after the Father (Infinite Love), I must be only the Word of God.

> Every instant I live the Word is a kiss upon the Lips of Jesus, those Lips which spoke only Words of Life.
>
> (*P'49* 195–196, 24 July 1949).

> . . . living the Word of Life in the will of God, moment by moment, I am the living Word, the living expression of love.
>
> (*P'49* 967, 9 November 1949).

When Lubich says that she "lives Jesus", she actually wants to make the reverse statement. It is no longer she who lives, but Jesus who lives in her—as the apostle Paul had expressed it in Galatians: "It is no longer I who live, but Christ who lives in me" (Gal 2:20). The believer no longer sees what he believes in front of himself—in Paul's case: God in Jesus Christ—but internalizes it to such an extent that he experiences it as the subject of his own life. This change is expressed in various places in *Paradise '49*, for example in a passage from August 28:

> Jesus in me is all me. And yet he continuously presses upon my humanity telling me that he wants to live!
>
> (*P'49* 477).

And again, on September 6:

> We truly live Jesus and Jesus lives us. That "Live me utterly, my Love!" which we would say to him, is done. And to him we say: "I live you utterly."
>
> (*P'49* 581).

On 8 December 1949, in a text summarizing the period that began in July, Lubich connects the image of the change of place (being taken into the "bosom of the Father") with that of the change in subject:

> It is the point where the created dies into the Uncreated, where nothingness is lost in the Bosom of the Father, where the Spirit pronounces with our lips: Abba-Father.
>
> Then our soul is the soul of Jesus.
>
> It is no longer we who live; it is Christ, truly, who lives in us.
>
> Then within the Bosom of the Father we come to know all the inhabitants of heaven and we understand the work God does in us, clothing us bit by bit in the divine.
>
> And this is what God revealed this summer.
>
> (*P'49* pp. 42–46, 8 December 1949).

## 5. Through the Other's Eyes: The Shift in Hermeneutics

### 5.1. The Shift to God as Love

For Lubich, truth of faith is "lived" truth, or existentially experienced truth. In this last chapter some texts from the corpus of *Paradise '49* regarding this point will be quoted more extensively. Truth is not primarily an assertion whose correctness can be shown dogmatically, but it is the life of God in man, which does, however, possess an impulse to express itself linguistically. The God in whom Lubich believes is in his essence love, and this love yearns to be believed, experienced, and communicated.

The connection between the conception of faith in God as love and Lubich's usage of "to live" and "to be" can be discerned in several passages. It can be shown, for example, in the text quoted above about Jesus who "wants to live". It is part of a section that can be classified among the *Realities*:

> Today we are Jesus. And Jesus in us walks among people and people do not know it. Our walking, doing, loving, smiling, sleeping is his and bears fruit for souls as was borne by his: loving, doing, smiling, and so on.
>
> I feel his Heart in mine and his Soul in mine. His Heart is infinite Love: Love, Love, Love … it cannot be said how much it is Love. And I feel Love for all people who do not have it and I will give it to all.
>
> And I feel his Soul in mine: because in me I feel Life for me and for everyone.
>
> Jesus in me is all me. And yet he continuously presses upon my humanity telling me that he wants to live!
>
> His Light is also in me: but now it is a Light, Love, Life. I do not know if it is more one than the other: I am living Paradise already here below: I am living together in an ever greater fullness with the comprehensors in beatitude.
>
> (*P'49* pp. 474–478, 25 August 1949).

As part of a *Reality*, this passage must be understood on the one hand in light of the eschatological dimension of the mystical experience of 1949. However, its meaning also goes beyond that incident because it points to pathways that are also found in other parts of *Paradise '49* that speak more generally of living by faith. Thus, the above passage is followed by another, from only a few days later, referring to this earthly reality and relating it to faith in God as love. Put in another way, God operates in the world in and through human persons:

> Lord, give me all the lonely ... I have felt in my heart the passion that fills your heart for all the forsakenness in which the whole world is drifting.
>
> I love every being that is sick and alone: even plants in distress cause me pain ... even animals left alone.
>
> Who consoles their weeping?
>
> Who mourns their slow death?
>
> And who clasps to their own the heart in despair?
>
> Grant me, my God, to be in this world the tangible sacrament of your Love, of your being Love: to be your arms that clasp to themselves and consume in love all the loneliness of the world.
>
> (*P'49* pp. 541–546, 1 September 1949).

As Lubich affirmed, the belief that an Other—God—is the real subject of one's life, leads to a change of perspective on the world that surrounds us. This is most clearly expressed in a text from *Paradise '49* entitled *Resurrezione di Roma*, which has been published with commentary on several occasions.[14] The metaphor of the eyes of God plays an important role in this text. Lubich imagines how Jesus must have looked at the world surrounding him two thousand years ago, and how he looks at the suffering and sin of the world today if we let him live in us now. She proposes how this world can find, through love, a life closer to its original destiny.

> And I make contact with the Fire that, invading the whole of my humanity given me by God, makes me another Christ, another God-who-is-human by participation, in such a way that my humanity merges with the divine and my eyes are no longer lifeless, but, through the pupil, which is an open space onto the soul, through which passes all the Light that is within (if I let God live in me), I look at the world and at things. But it is no longer I who look, it is Christ in me who looks and sees again the blind to enlighten, the mute to make speak, and the crippled to make walk—blind to the vision of God within and outside them; mute to the Word of God that also speaks within them and by them could be conveyed to their brothers and sisters to reawaken them to the Truth; the crippled unable to move because ignorant of the divine will that from the depths of their hearts spurs them to the eternal motion that is the eternal Love, where by conveying Fire one is set ablaze.
>
> In such a way that, opening my eyes again to what lies outside, I see humanity with the eye of God who believes all things because he is Love.
>
> (*P'49* pp. 717–718, October 1949).

The "eyes of God" clearly see and identify everything that is dark and negative in the world and also name it as such, but at the same time do not condemn the people affected by it, instead seeing in them the possibility of conversion and renewal. The eyes of God "read" the world in a specific, consciously chosen perspective characterized by love. This transformation represents a shift in hermeneutics, particularly in the perception and the acceptance of others (and those quite foreign to subject).

### 5.2. The Shift to God in the Other

This new perspective allows one to break out of the *incurvatio in se ipsum*, opening oneself to the fact that the God, in whom one believes and from whom one lives, is also present in the other. One thus comes to devote attention to "all the flowers" in the garden of God's creation, as Lubich writes in another already published text from one month later[15] which pointedly states that it is a matter of "losing the God in self for God in our brothers and sisters" (*P'49* p. 946, 6 November 1949).

In this way, a path is laid out over the course of *Paradise '49* which was not yet in view in 1949 but which gradually emerged over the course of Lubich's life and the history of the Focolare Movement: the path of ecumenical and interreligious dialogue. This was a

consequence of the understanding of God as expressed in Lubich's mystical writings in particular. These texts put forth the notion that opening oneself to the Other does not render one weak or vulnerable, but rather offers a chance to better understand one's own God. To look at the other with the eyes of God also entails refraining from a desire to possess:

> In however many neighbors you meet throughout your day, from morning to night, in all of them see Jesus.
>
> If your eye is simple, the one who looks through it is God. And God is Love, and love seeks to unite, winning over.
>
> How many, in error, look at people and at things in order to possess them. And their look is one of egoism or of envy or, in any case, of sin. Or they look within to possess themselves, to possess their own souls, and their look is lifeless because it is bored or troubled.
>
> The soul, because it is God's image, is love, and love turned in on itself is like a flame that, not fed, dies out.
>
> Look outside yourself, not in yourself, not in things, not in persons: look at God outside yourself to unite yourself with him.
>
> (*P'49* pp. 903–907, November 1949)[16].

Love means giving, means letting go, it means—to put it provocatively—letting oneself be possessed by the other, because this other does not lead us away from God but, on the contrary, leads us closer to Him:

> Let yourself be possessed by each one—out of love for Jesus; let yourself be "eaten" by them—like another Eucharist. Put yourself completely at their service, which is service to God, and your brother or sister will come to you and love you. And the fulfillment of God's every desire is fraternal love, which is a command: "I give you a new commandment, that you love one another."
>
> (*P'49* 915, November 1949).

Love removes every potential barrier separating the believer from God—it is concrete love for one's concrete neighbor. "To live" in God means "to live" in one's neighbor:

> There are many ways to clean a room: picking up one straw after another, using a little broom or a bigger one, a large vacuum cleaner, and so on. Or else, to have a clean place, we can change rooms and all is done.
>
> Likewise for our becoming holy.
>
> Rather than working a great deal, we can immediately step aside and let Jesus live in us.
>
> That is to say, living transferred into the Other: in our neighbor, for example, who, moment by moment, is close to us: living the other's life in all its fullness.
>
> Just as in the Trinity—and this alone is Love—the Father lives in the Son and vice versa. And their Love for one another is the Holy Spirit. When we live transferred into our brother or sister (you must lose your life to re-find it), as soon as we have to return into ourselves to respond to them, we find in ourselves a Third: the Holy Spirit, who has taken the place of our emptiness.
>
> Now, we can enter into the other in various ways: pushing ourselves in like someone big who wants to get in through a small door ... and this is how someone acts who does not listen to the very end (someone who does not die totally into the brother or sister who is Paradise for the self, the Kingdom for the self) and wants to give replies gathered bit by bit in his or her own head that may be inspired but are not that breath of the Holy Spirit which will give life to the other.
>
> There are those (passionate lovers of Jesus Forsaken) who more willingly die than live and who listen to their brother or sister all the way to the end, not

worried about the reply, which will be given in the end by the Holy Spirit who summarizes in a few words, or in one, all the medicine for that soul.

(*P'49* pp. 605–611, 8 September 1949).

To live "transferred into the other": in this formulation it becomes clear that for Lubich the encounter with God is inseparable from the encounter with fellow human beings, as both constitute the "other". To venture out into the encounter with the other is to create an open space, one which Lubich sees as filled with the Holy Spirit, as she puts it above. This faith in the Holy Spirit, in turn, is what takes away her concern about losing herself in this encounter with the other or about relativizing the truth of God from which she lives. In dialogue with believers of other religions, she never questioned or hid her rootedness in her own Christian tradition. Nevertheless, at the same time, she did not attempt to proselytize others. Mission was certainly an important topic for Lubich, but for her it was the mission of love and the conversion to love that opened the very space in which God himself could operate.

From Lubich's perspective, truth is first and foremost lived and experienced, but it always calls for communication, which can only be understood and expressed through the use of language. The language of faith and thus also the language of theology are indispensable to a Christian conception of God in which the word and communication are central. The Word, rather than silence, is the culmination of the encounter with God. Nonetheless, the necessity of communication in an understanding of faith that places love centrally leaves open the possibility that other linguistic and cultural forms of expression may fulfil an analogous function. Interreligious dialogue is thus the way to learn, and relearn, that the divine is beyond our powers of description:

It is necessary to put ourselves before everyone in an attitude of learning, for we really have something to learn.

(*P'49* p. 540, 28 August 1949).

The willingness to learn applies to every stage in the interreligious dialogue as Lubich understands it and as described in the introduction to this paper.[17] It involves, first of all, perceiving the concrete other person as a neighbor and taking him or her seriously. Based on this, an exchange of spiritual experiences can develop, in which each interlocutor can listen to the way the other lives from his faith. The aforementioned encounter with Warith Deen Mohammed was characterized by this (Coda 1997; Lemarié 2014; Mokrani 2008). In a further step, this can then also lead to a dialogue on an academic level, which takes up classical controversial points in the dialogue of religions and is characterized by the willingness to deepen one's own understanding of the truth.[18] However, this academic dialogue is always supported and inspired by the other two levels which must never be missing.

This perspective renders interreligious dialogue honest and open at the same time. It is not a matter of talking about mysticism, but of living out of it: for Chiara Lubich and the Focolare Movement, the experiences of the years 1949 and 1950 were a crucial milestone in the formation of this open understanding of faith, and they eventually came to light in *Paradise '49*. This corpus, which will hopefully soon be published in its entirety, could, therefore, still serve as a fount of inspiration for dialogue and for theological thinking in general.

**Funding:** This research was supported by a Hasso Plattner Excellence Research Grant (LBUS-HPI-ERG-2020-XX), financed the Knowledge Transfer Center of the Lucian Blaga University of Sibiu.

**Institutional Review Board Statement:** Not applicable.

**Informed Consent Statement:** Not applicable.

**Data Availability Statement:** Not applicable.

**Conflicts of Interest:** The author declares no conflict of interest.

## Notes

1. The tension between the prevailing Catholic culture of the time and the charismatic personality of Chiara Lubich is well described in a sociological doctoral thesis (Callebaut 2017).

2. The 1962 statutes were later substantially revised and adopted in their current form in 1990, with amendments in 1994 and 1998.

3. The first publication appeared in 1959 (Lubich 1959). A collection of important spiritual texts is found in the four volumes of *Scritti Spirituali* (Lubich 1978, 1979, 1980, 1981). A systematic summary in the context of a theological analysis is offered by the author of this paper (Tobler 2002, pp. 93–188).

4. At the *Sophia University Institute* (Loppiano/Florence), Michel Bronzwaer is working on a dissertation on all of Lubich's texts written before 1949 which have been preserved, which are mainly letters. The publication of this work will provide more detailed information.

5. The historical context and people involved are described in (Abignente and Delama 2019, pp. 77–90).

6. Nuova Umanità. Rivista bimestrale di cultura, Città Nuova: Roma, 1978ff.

7. Studi della Scuola Abbà, Città Nuova: Roma 2012ff.

8. See: https://docs.lib.purdue.edu/claritas/ accessed on 30 May 2022.

9. Lubich (2004). The sections of this manuscript are numbered from 1 to 1724, and the footnotes are numbered from 1 to 1078. In previous publications, the method of using the section number rather than the page number for a citation has proven effective, along with the associated date if cited.

10. Lubich herself uses this appellation: Lubich (2004, footnote 364 to para. 426).

11. This description was written in retrospect in 1986, but there is also older written evidence, dated between August and December 1949, which attests to these early days.

12. This understanding of theology is taken up and elaborated by Stefan Ulz (Ulz 2019, pp. 143–52).

13. *P'49*, author's comment before para. 434 of 19 August 1949.

14. As the only fragment from *Paradise '49,* it was already published at that time in publisher Igino Giordani's journal (La Via 1 (36/1949) 5) and, probably also for this reason, it has the style of a self-contained, freestanding text. It was then printed again, in its full extent for the first time, in 1995 (in: *Nuova Umanità* no. 102, 17 (1995) pp. 5–8). It was recently subject of a comprehensive analysis in a volume of *Scuola Abbà* (Blaumeiser and Rossi 2017).

15. *P'49* pp. 931–947, 6 November 1949. It is entitled "Guardare tutti i fiori" and was also analyzed in detail in a volume published by *Scuola Abbà* (Abignente et al. 2014).

16. This excerpt is part of a longer passage (*P'49* pp. 903–924) written as a commentary on the Bible verse Lk 11:34. Parts of it were occasionally quoted, but as a complete text it first appeared in German translation in: Tobler (2021).

17. A description together with a theological analysis of the relationships with representatives of different world religions can be found in (Catalano 2022).

18. This was aptly expressed by the sociologist of religion Peter L. Berger when he speaks about "dialogic engagement" as the most valuable kind of interreligious dialogue (Berger and Schweitzer 2011). Examples based on the spirituality of Chiara Lubich can be found in Chapter 7 of Mitchell (2020, pp. 451–531) which is dedicated to "Implications of *Paradise '49* for Interreligious Dialogue", with five comparative studies mainly about the Buddhist–Christian relationship.

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
