# Peer review of "Reading through the Other’s Eyes: The Mystical Foundations of Interreligious Dialogue in Chiara Lubich’s Paradise ‘49"

_religions, doi:10.3390/rel13070638_

Round 1
Reviewer 1 Report
Reading through the Other's Eyes: The mystical foundations of interreligious dialogue in Chiara Lubich’s Paradise '49
Review for the Journal of Religion.
Form. The paper is well written and documented. Clear in structure and content. Title well-chosen covering content. Primary source used for analysis. Scientifically grounded analysis performed. The selection of fragments needs some more explanation (illustrations of text? proof of arguments?). Conclusions seem to be supported by the analysis, but need a better tie for avoiding a jump. They look more like an inference than a conclusion now (weak point of a strong paper).
Content. The content of the paper suits the Journal and the theme of this issue.
Some questions and remarks:
Line 41. Please define “dialogue”. I think the author does not refer to a cognitive exchange of arguments, but rather to a shared mystical experience. The word is introduced carefully, but with respect to the title of the paper deserves further determination(s).
Line 70. It actually looks like there are two questions posed here not just one. I would prefer to join them into one question that looks like the second one as stated.
Line 123. These general characteristics of mysticism surely serve the purpose of the paper and I like the addition of language to Tobler’s frame of reference. I miss the aspect of aesthetics as (often mentioned) aspect of mysticism. See e.g. line 423 where Lubich speaks of “comprehensors in beatitude”. Could the author elaborate on that also in relation to language?
Line 208. Year 2004? Typo? Lubich died in 2008.
Line 254. The citation of Lubich needs some clarification. Who is “us”? What is the meaning of this remarkable way of writing?
Line 267. I agree with the interpretation of “to live” as understanding, but would think it is an understanding in a practical way, a practice like e.g. Pierre Hadot (2010) points out. What do the authors think of that?
Line 284-285. Not so clear. What does Coda mean here by stating God is not the object of theology? It looks like a deviation from the general/classical definition of Theology. I suppose this is the modification of defining theology by Chiara Lubich.
Line 312. This seems to be the answer to my question about line 254. Consider referring to it there.
Line 357. “..this indicates” needs some clarification. Why or how can you infer meaning by the frequency of occurrence of words? (more general methodological)
Line 548-555. This seems to be a jump to conclusions. I can follow the need of “the other” in Lubich’s way of live, but not the need of a (linguistic) dialogue. A meeting might suffice (or not?). Could the author elaborate on this. The author is close to this conclusion, but maybe could reformulate this paragraph. What kind of dialogue is meant? A meeting? A mystical exchange? I wonder what the author thinks.
Line 560. Again (see also remark at line 41). What kind of dialogue is meant here? I think the conclusion of the paper should also hold a definition of dialogue from Lubich’s point of view and point out the role of mysticism in the dialogue between religions. The argument seems to state the dialogue may stem from mysticism as the example of Lubich shows. Is there also a role for mysticism in this dialogue? For example, did Lubich talked about mysticism in the Malcolm X Mosque and was this the reason for the lasting relationship between her and their Iman? I would like the author to elaborate on that in order to complete their line of reasoning with a conclusion instead with some kind of inference.
In general, with respect to content. Well-developed argument. Clear line of reasoning. Conclusion could be strengthened.

Reviewer 2 Report
The text is interesting but needs some additions and clarifications.
1. The title of the article and the purpose stated in lines 68-73 are quite distinct from each other.
2.Given the title of the article, it must be said that the introduction lacks an explanation of what interreligious dialogue is. This is necessary if one wants to write what are the mystical foundations of interreligious dialogue. The teaching of Paul VI and the Second Vatican Council in this regard cannot be omitted, even if only in a few sentences. The author mentions Pope John Paul II in line 34. It is difficult to omit him when presenting the mark of his pontificate in the development of interreligious dialogue. Having in mind at least the meetings in Assisi.
3. In line 68 we read that we are dealing with an 'essay'. As we know, Religions does not publish essays, but academic articles.
4. There is no title above the table and no source below it.
5. The text lacks a discussion. It would be useful to clarify in it what, then, are the mystical foundations of interreligious dialogue contained in Paradise'49 and what relevance they have for interreligious dialogue.
6. It should also be remembered that there is, for example, a literature widely available on mysticism and interreligious dialogue on the Internet. See e.g.:
K. Waaijman Mystical Perspectives in Interreligious Dialogue; Mohammad Saeedimehr, Islamic Mysticism and Interreligious Dialogue,
as well as many other texts that can be helpful for discussion and conclusions.
Round 2
Reviewer 2 Report
I do not fully agree with the arguments of the author of the text.
However, this is not a reason for the article not to be accepted for publication in Religions.